# Addressing the Real-World Challenges of Immunoresistance to Botulinum Neurotoxin A in Aesthetic Practice: Insights and Recommendations from a Panel Discussion in Hong Kong

**DOI:** 10.3390/toxins15070456

**Published:** 2023-07-12

**Authors:** Wilson W. S. Ho, Lisa Chan, Niamh Corduff, Wang-Tak Lau, Michael U. Martin, Clifton Ming Tay, Sandy Wang, Raymond Wu

**Affiliations:** 1The Specialists: Lasers, Aesthetic and Plastic Surgery, Central, Hong Kong; 2EverKeen Medical Centre, Tin Hau, Hong Kong; drchan.everkeen@gmail.com; 3Cosmetic Refinement Clinic, Geelong, VIC 3220, Australia; niamh.crc@outlook.com; 4Independent Researcher, Kowloon, Hong Kong; lauwtt@yahoo.com; 5Independent Researcher, 31832 Springe, Germany; michael.u.martin@web.de; 6Merz Asia Pacific Pte., Ltd., Singapore 138567, Singapore; clifton.tay@merz.sg; 7Independent Researcher, Causeway Bay, Hong Kong; iswang616@hotmail.com; 8Asia Pacific Aesthetic Academy, Kowloon, Hong Kong; wu.raymond@gmail.com

**Keywords:** botulinum neurotoxin A, immunoresistance, secondary non-response, neutralizing antibodies

## Abstract

With increasing off-label aesthetic indications using higher botulinum neurotoxin A (BoNT-A) doses and individuals starting treatment at a younger age, particularly in Asia, there is a greater risk of developing immunoresistance to BoNT-A. This warrants more in-depth discussions by aesthetic practitioners to inform patients and guide shared decision-making. A panel comprising international experts and experienced aesthetic practitioners in Hong Kong discussed the implications and impact of immunoresistance to BoNT-A in contemporary aesthetic practice, along with practical strategies for risk management. Following discussions on a clinical case example and the results of an Asia-Pacific consumer study, the panel concurred that it is a priority to raise awareness of the possibility and long-term implications of secondary non-response due to immunoresistance to BoNT-A. Where efficacy and safety are comparable, a formulation with the lowest immunogenicity is preferred. The panel also strongly favored a thorough initial consultation to establish the patient’s treatment history, explain treatment side effects, including the causes and consequences of immunoresistance, and discuss treatment goals. Patients look to aesthetic practitioners for guidance, placing an important responsibility on practitioners to adopt risk-mitigating strategies and adequately communicate important risks to patients to support informed and prudent BoNT-A treatment decisions.

## 1. Introduction

Botulinum neurotoxin A (BoNT-A) injection is the most frequently performed aesthetic procedure worldwide [1,2,3], with diverse applications that leverage its transient neuromodulatory activity [4,5]. Standard BoNT-A aesthetic applications involve forehead, glabella, and crow’s feet indications, but off-label applications such as intradermal injection and body contouring, which require much higher doses, have increased steadily, especially in Asia [6,7,8,9]. Due to the limited duration of action of BoNT-A neuromodulation, repeated injections are necessary to maintain treatment effects. Thus, as with any injection of therapeutic proteins, the risk of developing neutralizing antibodies resulting in immunoresistance and loss of response to BoNT-A should be considered [10,11,12].

Presently, onabotulinumtoxinA (ONA), abobotulinumtoxinA (ABO), and incobotulinumtoxinA (INCO) are the most widely used BoNT-A formulations, and all three are United States Food and Drug Administration (FDA) approved for medical and aesthetic indications [13,14,15]. In recent years, many newer formulations, predominantly from Korea, have become available. Although all formulations contain the biologically active 150 kDa neurotoxin, only the INCO formulation is purified to eliminate inactive neurotoxin and pharmacologically unnecessary impurities [5,11,13]. In contrast, ABO and ONA contain complexing proteins and other non-functional impurities [5,11,13,16].

These pharmacologically unnecessary impurities include bacteria-derived components such as flagellin and bacterial DNA. Unlike the pure bioactive 150 kDa BoNT-A molecule, these can act as immune adjuvants to stimulate an adaptive immune response and the production of neutralizing antibodies (NAbs) against BoNT-A and other injected proteins. BoNT-A-directed NAbs can contribute to complete or partial loss of neuromodulatory effects, termed NAb-related secondary non-response (SNR) [16,17]. Individuals with SNR initially had a good therapeutic response to BoNT-A treatment but later experienced partial or complete loss of response with subsequent injections. Clinical signs associated with partial SNR (PSNR) include dose and interval creep, where higher BoNT-A doses or more frequent injections are needed to achieve the same level of therapeutic effects as before [11]. For patients with complete SNR (CSNR), there is a complete loss of response and clinical effects, even if higher BoNT-A doses are injected [18]. To address SNR appropriately, it is important for practitioners to distinguish BoNT-A NAb-related SNR from other possible causes, including insufficient toxin dose, improper injection technique, and inappropriate target muscle selection [11].

In this context, two trends in the aesthetic use of BoNT-A warrant special mention. The expanding number of off-label aesthetic applications that use higher BoNT-A doses (approaching those for medical BoNT-A use) is well known [6,7,8,19,20]. Additionally, more individuals are receiving BoNT-A treatments, often starting at a younger age; this may increase their overall lifetime exposure and thus the risk of developing immunoresistance to BoNT-A [8,21,22,23]. It is therefore pertinent for aesthetic practitioners and other relevant stakeholders to be aware of the long-term implications of BoNT-A treatment patterns so as to guide informed decision-making with patients.

A round-table discussion in Hong Kong, involving a panel of nine local aesthetic practitioners and international experts, was held to discuss the implications of immunoresistance to BoNT-A in contemporary aesthetic practice and practical strategies for risk management. The panel discussed published clinical evidence on NAb-related SNR with aesthetic BoNT-A treatment, real-world data from an Asia-Pacific consumer study on BoNT-A user perceptions and experiences of diminishing efficacy, and a patient case illustrating the development of NAb-related immunoresistance. Panel members shared their perceptions and experiences with immunoresistance to BoNT-A. They also discussed ways to raise awareness of the issue locally and strategies to mitigate the risk of immunoresistance to BoNT-A in everyday practice.

## 2. Results of the Panel Discussion

### 2.1. Clinical and Real-World Evidence on Immunoresistance to BoNT-A in Aesthetic Practice

The panel discussed the challenges of drawing conclusions about the extent and impact of immunoresistance to BoNT-A in aesthetic applications owing to the paucity of published research. According to published estimates from systematic reviews and meta-analyses (SRMAs), rates of immunoresistance to BoNT-A in aesthetics range from 0.2–0.4%, much lower than in therapeutic applications [5]. However, it was emphasized that these estimates were based on data from short-term randomized controlled trials and observational studies for a few on-label applications, and data from case reports or case series were excluded from these SRMAs. A recent literature review that included case reports/series in the literature search identified 13 cases of NAb-related SNR from aesthetic BoNT-A treatment that were excluded from published SRMAs [5]. Although published case reports on immunoresistance cannot be used to estimate rates of immunoresistance as they cover a wider range of off-label aesthetic BoNT-A applications, these individual case narratives clearly indicate patterns of diminishing efficacy and loss of response over time with repeated BoNT-A treatments, even before immunoresistance was confirmed [5].

Furthermore, the panel discussed analyses of data from the US FDA Adverse Event Reporting System (FAERS) database on diminishing efficacy with aesthetic BoNT-A treatments between March 2014 and June 2019. This FAERS analysis identified 23,789 reports of diminishing efficacy [24,25]. Most of the reports in the dataset were associated with users of the ONA (84.6% of reports) and ABO (10.2%) formulations; 5.3% of reports were associated with INCO users. It is important to note that these are unadjusted statistics that include patients with exposure to more than one BoNT-A formulation and that do not take into account the relative market share of these formulations. Consequently, the data might not reflect the true incidence of diminishing efficacy with the exclusive use of these formulations. As others have previously noted, among patients who were treated with BoNT-A for therapeutic indications, there have been no reported cases of NAb-related SNR in patients who exclusively used the INCO formulation, in contrast to patients who exclusively used the ONA (0.6%) or ABO (5.3%) formulations [5,18]. For ONA and ABO, diminishing efficacy was more frequently reported in those who had received treatment for more than one year versus less than one year (ONA: 14.4% versus 7.6%, *p* < 0.001; ABO: 9.9% versus 3.3%, *p* < 0.001) [24,25]. This finding is consistent with the risk of NAb-related SNR increasing with longer exposure to BoNT-A. For patients who received INCO, there was no significant difference in the relative incidence of diminishing efficacy for more than one year of treatment versus less than one year of treatment (0.0% versus 3.7%, *p* = 0.62). This suggests that the cases of diminishing efficacy reported in INCO users were more likely to be related to factors other than NAb, e.g., incorrect injection technique. The researchers also noted that BoNT-A NAb formation was unlikely to account for all reported cases of diminished efficacy. Nonetheless, results from these two studies support the notion of a higher real-world prevalence of immunoresistance to BoNT-A than would be assumed based on published studies.

### 2.2. Consumer Awareness and Perspectives on Diminishing BoNT-A Treatment Efficacy over Time

As noted above, clinical studies have not captured relevant long-term data on aesthetic patients’ experiences of SNR after BoNT-A, such as gradually diminishing efficacy after multiple treatments. To explore the potential extent, impact, and awareness of immunoresistance to BoNT-A from the aesthetic patient’s perspective, survey-based studies were conducted in the Asia-Pacific region in 2018 (six countries/territories) and in 2021 (eight countries/territories). These surveys utilized similar methodology and respondent screening criteria and included consumers aged 21–55 who had received at least three prior aesthetic treatments with BoNT-A. An overview of the survey methods is presented in the Methods section below, and further details are presented in Appendix A.

Overall, 2201 and 2441 respondents completed the survey in 2018 and 2021, respectively. In both the 2018 and 2021 surveys, most respondents were female (85%), and the average age was 37 years. The demographics and key characteristics of the 2018 and 2021 survey respondents are summarized in Appendix A.

Most respondents reported being aware of the concept of diminishing treatment efficacy (2018: 81%; 2021: 87%) (Figure 1A). When respondents were asked about the signs/symptoms that they would associate with diminishing treatment efficacy, the most frequently mentioned signs were a shorter duration of treatment effect (2018: 80%; 2021: 69%) and a weaker effect (less than the desired outcome) after treatment (2018: 47%; 2021: 55%) compared with previous treatments. In addition to these, respondents also associated a lack of response after treatment (2018: 24%; 2021: 37%) and using higher doses to achieve the same treatment outcome (2018: not assessed; 2021: 29%) with the concept of diminishing efficacy, albeit less frequently. Strikingly, well over half of respondents (69% in 2018; 79% in 2021) reported they had experienced diminishing efficacy (Figure 1B).

The results also illustrated the emotional impact of diminishing efficacy on respondents. More than half reported feeling disappointed, and about one-third reported feeling anxious, sad, or frustrated (Appendix A). Of note, the considerable emotional impact that most respondents reported did not markedly affect their willingness to continue receiving BoNT-A treatment. In fact, most respondents who experienced diminishing efficacy reported continuing with their treatment (2018: 64%; 2021: 70%) despite the experience. Consistently in both surveys, almost all respondents who continued BoNT-A treatment even after experiencing diminished efficacy (2018: 97%; 2021: 95%) reported that they did so to maintain or retain their looks.

Next, the surveys explored the depth of respondents’ awareness and understanding of BoNT-A NAb development and its implications, especially diminished treatment efficacy. The overall proportion of respondents who indicated that they were aware of “toxin resistance” due to BoNT-A NAb formation was higher in 2021 (71%) than in 2018 (59%). Of note, even though the majority of respondents (2021: 80%; 2018: 72%) indicated that they were very or somewhat worried that they might develop NAbs to BoNT-A, a much lower proportion (2021: 60%; 2018: 36%) were aware that the presence of impurities such as complexing proteins could increase the risk of NAb development.

The findings also suggest that most aesthetic patients want to be adequately informed of the risk factors for developing immunoresistance and that it is important to them that such information is accurately conveyed. In both the 2018 and 2021 surveys, over 80% of respondents who said they were aware of “toxin resistance” also indicated that they had discussed the risks of NAb formation with their HCPs (84% and 89%, respectively). However, in light of the limited awareness of factors such as complexing proteins and other impurities that can trigger an immune response leading to NAb formation, the panel considered that HCPs need to ensure they communicate both the causes and the consequences of “toxin resistance” in ways that are easy for a layperson to understand.

In the 2021 survey, respondents who decided to continue BoNT-A treatment after experiencing diminishing efficacy tended to adopt one of a few approaches: use of higher BoNT-A doses than in previous treatments (43%), treating at shorter intervals (31%), or switching to a different formulation (16%). Although increasing BoNT-A doses and shortening treatment intervals may produce the desired effects in the short term, these approaches are unlikely to be adequate long-term solutions for individuals who are experiencing a loss of efficacy due to NAb formation. Thus, it is important for HCPs to investigate the possible causes of diminished efficacy in each patient before recommending changes to treatment. In addition, these treatment-seeking behaviors suggest that some aesthetic patients might not perceive the connection between NAb formation and the diminishing efficacy they are experiencing, despite their worries about the issue of NAb formation. This further highlights the importance of clear communication and education on immunoresistance by HCPs.

The panel noted that the trends for Hong Kong appeared consistent with those identified for the overall Asia-Pacific dataset. For example, of the respondents in the Hong Kong samples (*n* = 250 in 2018, *n* = 255 in 2021), nearly three-quarters reported that they had experienced diminishing efficacy (2018: 72%; 2021: 76%) (Appendix A). There were also moderately high levels of worry among Hong Kong respondents about developing NAbs against BoNT-A (2018: 81%; 2021: 77%); yet, only 55% of respondents in 2021 (30% in 2018) knew about key factors that could lead to NAb development. Hong Kong respondents likewise experienced the negative emotional impact of diminishing efficacy, such as feelings of disappointment, which were reported by over half of the respondents (2018: 54%; 2021: 52%) (Appendix A). Despite the negative emotional impact of diminishing treatment efficacy, more than two-thirds of the respondents (2018: 68%; 2021: 71%) nevertheless chose to continue with BoNT-A treatment, akin to the trend observed in the overall Asia-Pacific sample.

### 2.3. Patient Case Discussion

Individual case narratives offer critical insights into the “natural history” of BoNT-A-related SNR as it develops and manifests clinically over the course of multiple aesthetic treatments [10,26]. As an example, the case of a patient in Hong Kong who developed complete NAb-related SNR after multiple BoNT-A treatments for body sculpting (six treatments) and facial indications (four treatments) (Figure 2) was presented at the meeting.

The patient was a female who started her BoNT-A treatment journey in her 20 s. She reported receiving her first BoNT-A treatment (50–100 U per calf, ONA) for calf sculpting in 2014, then two further treatments for calf sculpting (same dose at 6-month intervals) until late 2015, when she decided to take additional treatments for facial indications (forehead, glabella, masseter, total dose of approximately 100 U). Subsequently, further treatments for both calf sculpting and facial indications were administered 9–12 months apart. In early 2017, after her sixth treatment for calf sculpting and fourth treatment for facial indications, the patient noticed a reduction in treatment effects even though she was injected with similar doses as previous treatments, and that the effects did not last as long as before, indicating PSNR.

Despite these signs, she reported receiving an additional 3–4 BoNT-A treatments from different injectors for the same indications between 2018 and 2020. The formulations used for these treatments included ABO, ONA, and INCO. Despite switching formulations, the patient continued to experience a gradual decline in clinical effects with successive treatments, finally experiencing a complete loss of clinical effects, indicating CSNR. In 2020, when she went for another BoNT-A injection, the patient discussed with the HCP her experience of lack of effects in previous treatments. The HCP suggested NAb formation as a possible cause of SNR and advised her to avoid further treatments for some time if she still showed little or no response to this treatment. In February 2021, the patient sought a second opinion, and her serum sample was sent for NAb testing (mouse hemi-diaphragm assay [MHDA], Toxogen, Hannover, Germany). The result was highly positive for NAbs against BoNT-A. She has not received any BoNT-A treatments since. 

The patient reported feeling shocked that the BoNT-A treatment stopped working after only a few treatments and wishing that she had been better informed of the risks before initiating aesthetic BoNT-A treatment. She also expressed worry about the potential implications for her health if she were to need BoNT-A for medical treatment in the future. This is in concordance with the negative emotional impact that respondents in the consumer survey reported when they experienced diminishing efficacy. With the considerable negative emotional impact that the experience of diminishing efficacy has on patients, it is vital for aesthetic practitioners to take precautions to minimize the risks of diminishing efficacy developing, thereby enhancing the patients’ overall experience during their BoNT-A treatment journey.

The BoNT-A treatment history presented here (BoNT-A doses, treatment intervals, number of treatments) was reconstructed based on the patient’s recall of events since detailed records from each clinic where she was treated were not available. The panel discussed the difficulties in establishing whether and why a patient may be experiencing diminishing BoNT-A efficacy in a timely fashion. As the panel noted, one contributing factor to difficulties in follow-up and documentation is the practice of seeking treatment at multiple clinics by aesthetic patients, especially when dissatisfied with treatment outcomes. For the above case, the possibility of immunoresistance to BoNT-A was only raised three years after the patient first noticed signs of diminishing efficacy, having received treatments from multiple clinics.

## 3. Discussion

### 3.1. Treatment Considerations for Best Clinical Practice

The panel drew on the consumer study data and patient case as a starting point to talk about preferred strategies for managing the risk of immunoresistance to BoNT-A in their own practice.

#### 3.1.1. Be Aware of the Possibility and Long-Term Implications of BoNT-A Immunoresistance

Considering the current BoNT-A treatment landscape, the panel agreed that it is critical for practitioners to recognize immunogenicity as a potential complication arising from continued BoNT-A use and to be vigilant in picking up subtle clinical signs that a patient might be developing immunoresistance (Figure 3). It was highlighted that appropriate clinical assessments and prudent treatment decisions could support continued BoNT-A use for aesthetic and medical indications with safe and satisfactory outcomes.

#### 3.1.2. Optimize Patient Journey and Consultation Process

The panel agreed that it is often challenging to establish an aesthetic patient’s treatment history. They noted that poorly documented treatment records are not unusual in their experience for patients who have received treatments for multiple indications at different clinics. This makes it challenging for HCPs to recognize diminishing BoNT-A efficacy and determine the underlying causes, which may contribute to under-reporting or missed diagnosis of NAb-related SNR. Thus, the panel emphasized the benefits of spending more time in consultation with patients to establish treatment histories, to educate patients on side effects of treatment, including the risk of immunoresistance, and understand patients’ individual treatment goals prior to administering treatment (Figure 3).

#### 3.1.3. Design Treatment Plan: Toxin Choice Is Important

A recent survey of Korean dermatologists reported that many considered cost-effectiveness as more important than the risk of NAb formation [8], highlighting the continued perception that immunoresistance to BoNT-A is not of substantive clinical concern in aesthetic settings. One panel member pointed out that although the overall reported prevalence of NAb-related SNR in published studies may be low, there appear to be relevant differences among formulations in terms of immunogenicity [5]. For example, there have been no reported cases of NAb formation in patients treated exclusively with pure BoNT-A formulations, such as INCO [27,28].

The panel discussed other considerations that influence the choice of toxin formulations in aesthetics, including real and perceived efficacy. One panel member highlighted that it is important to be aware that different formulations have similar efficacy but also subtly different characteristics, including their extent of spread and characteristic sensory symptoms (proprioception), that do not affect efficacy or longevity. For example, although treatment with INCO is associated with minimal sensations (such as stiffness and feeling frozen) that patients may be accustomed to with other formulations [29], INCO has efficacy and longevity comparable with other formulations [28,29,30]. Other panel members concurred and added that failing to advise patients on what to expect before switching formulations may lead to misperceptions that the treatment is not working well or that its effect has worn off prematurely.

Formulations were also reported to differ in terms of other characteristics, such as the extent of spread. Thus, practitioners should ensure proper injection techniques are used to get the desired results from each formulation. One study reported that ABO has a bigger spread compared with INCO and ONA when measuring the area of sweat inhibition after intramuscular injection [31]. This was echoed by various panel members, who noted that, in their experience, INCO has a smaller area of spread compared with ONA and ABO. Thus, more injection points for INCO are required and should be spaced closer together (without any change in total dose), especially for larger muscles such as the frontalis and masseter. This ensures that the motor endplates are sufficiently covered to achieve more precise and predictable results. The panel agreed that, where efficacy and safety are comparable, a formulation known to have low immunogenicity would be their first choice [32] (Figure 3). For treatment-naïve patients, the panel concurred that using a BoNT-A formulation with the lowest immunogenicity from the beginning will help to minimize the risk of immunoresistance and is therefore a prudent choice. They also emphasized the relevance of communicating with patients about the expected effects of treatment, including potential sensory differences with different formulations. “Before and after” photos were suggested as a way to reassure patients that their treatment is still effective even if they do not feel the same sensations as with other formulations.

#### 3.1.4. Recognize and Deal with Signs of Diminishing Efficacy

There are several diagnostic tests available to determine the presence of NAbs. The gold-standard diagnostic test is the MHDA [33]. However, due to the need for specialized laboratories and high costs for transporting samples, the MHDA is rarely accessed in aesthetic practice [11,34] (Figure 3). The panel emphasized that, with limited access, local practitioners have to rely on vigilance and clinical suspicion and not overlook subtle clinical signs of dose and interval “creep” that suggest a patient might have developed immunoresistance. Due to the high costs, the panel preferred to first utilize lower-cost and more accessible alternatives to screen for suspected immunoresistance to BoNT-A, such as enzyme-linked immunosorbent assays (ELISA) and clinical tests such as the frontalis test [35,36]. The MHDA can subsequently be used to confirm the presence of NAbs against BoNT-A. One panel member highlighted that as this decision involves the patient’s choice and preferences (e.g., the cost of antibody testing, the consequences of the frontalis test), it is important that practitioners establish adequate rapport and trust with the patient by being transparent when discussing available options. This transparency can help alleviate the emotional impact of SNR and improve the overall treatment experience so as to encourage patient retention.

### 3.2. The Role of Healthcare Professionals

The panel agreed that while all stakeholders, including professional societies, manufacturers, and regulators, can help promote awareness of immunoresistance, HCPs remain the primary trusted source of information and education for most patients.

The consumer study results suggest increasing awareness of the risk of developing NAbs among BoNT-A users. Over 90% of respondents in the 2021 survey expressed a preference for being informed of BoNT-A-related risks before initiating treatment. However, only 71% of respondents reported having been informed by their doctors, signifying an addressable gap in patient education. Thus, the panel strongly advocated spending more time, especially during initial consultations, to discuss the risks and options available prior to initiating BoNT-A treatment. For Hong Kong, since in-clinic beauty consultants are often the first point of contact for aesthetic patients, the panel also emphasized that local aesthetic clinics should take steps to ensure that their support staff are well informed of the causes and risks of immunoresistance to BoNT-A and are aware of best practices to help improve the overall patient experience.

## 4. Conclusions

Trends in contemporary aesthetic practice indicate increasing numbers of individuals are undergoing BoNT-A aesthetic treatment, often starting at a younger age and/or involving higher-dose applications. As individuals may continue receiving BoNT-A treatments on a long-term basis, it is important to adequately communicate important risks such as immunoresistance, which may contribute to loss of efficacy and options for future therapeutic use. Real-world observations of diminishing efficacy among experienced BoNT-A users serve to highlight that the risk of immunoresistance to BoNT-A is not trivial and should not be overlooked. Consumers are increasingly aware of immunoresistance but look to HCPs for guidance on managing treatment risks while attaining their desired aesthetic outcomes. Therefore, HCPs have a professional responsibility to adopt risk-mitigating strategies, starting with adequate discussions on immunoresistance to BoNT-A before initiating treatment, the choice of formulations, and expected treatment effects, so as to guide patients in making informed and prudent decisions about BoNT-A treatment.

## 5. Materials and Methods

### 5.1. Consumer Study

Survey-based studies were conducted in the Asia-Pacific region in 2018 (six countries) and in 2021 (eight countries) to explore the potential extent, impact, and awareness of immunoresistance to BoNT-A from the aesthetic patient’s perspective. These were consumer research panel-based online surveys using a semi-structured quantitative questionnaire. The surveys were carried out in accordance with locally applicable codes of conduct for consumer and market research by an independent research agency. Respondents were recruited from a consumer research panel managed by an established panel provider. Respondents’ participation in such panels is voluntary; the panel platform enables data collection using online surveys while protecting respondents’ anonymity and personal data privacy. As the recruitment process did not involve any hospital, physician, or institutional referrals, institutional review was not applicable. From consumer panels selected to be representative of each country, panel participants who met the following key criteria were selected: females or males aged between 21 and 55 years and received at least three BoNT-A treatments. Informed consent was obtained from each respondent before beginning the survey. Respondents were informed of the overall survey topic (aesthetic botulinum toxin use) but were blinded to the identity of the research sponsor (Merz Aesthetics) to reduce response bias. Appendix A provides further details of the data collection and analysis methodology.

### 5.2. Panel Discussion

A panel comprising international experts and experienced aesthetic practitioners was convened in Hong Kong. The panel, which was moderated by authors W.H. and C.T. and included all authors of this paper, discussed the published real-world evidence on BoNT-A immunoresistance and findings from the consumer study.

## Figures and Tables

**Figure 1 toxins-15-00456-f001:**
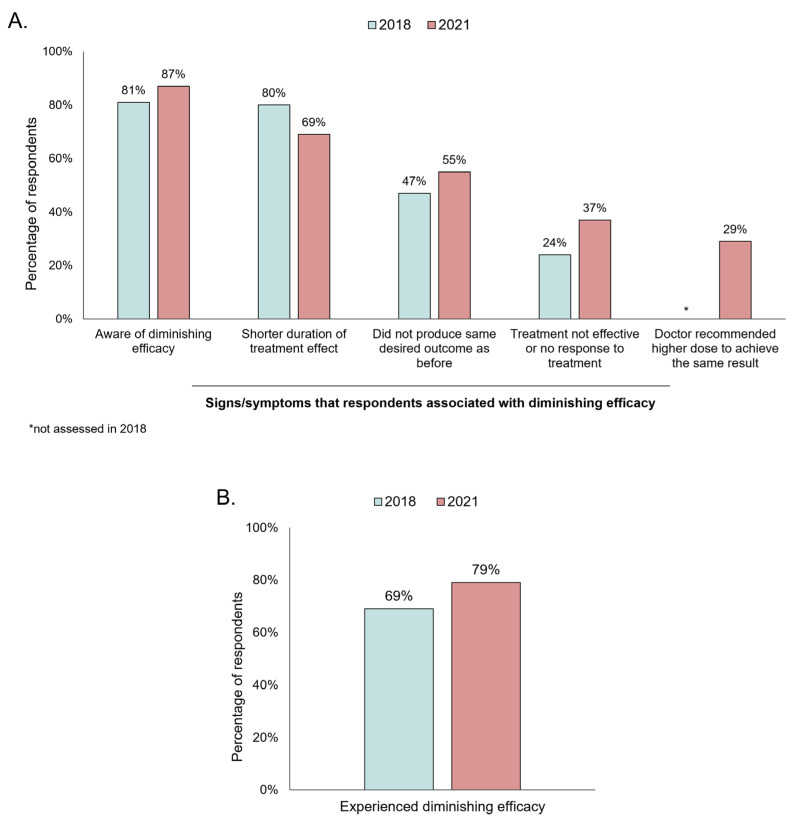
Consumer awareness and experience of diminishing BoNT-A treatment efficacy. (**A**) Awareness of diminishing efficacy and associated signs/symptoms. (**B**) Experienced diminishing efficacy.

**Figure 2 toxins-15-00456-f002:**
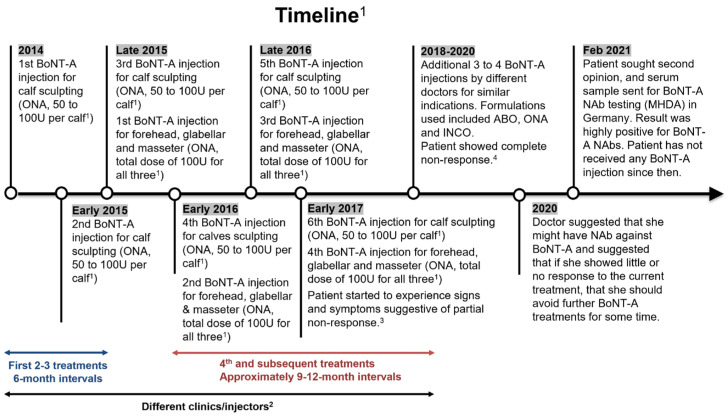
Patient’s BoNT-A treatment journey from first injection to positive test for NAb. BoNT-A, botulinum neurotoxin A; ABO, abobotulinumtoxinA; INCO, incobotulinumtoxinA; MHDA, mouse hemi-diaphragm assay; NAb, neutralizing antibody; ONA, onabotulinumtoxinA. ^1^ Details are based on patient’s recall; ^2^ The patient visited different clinics/injectors for each treatment; ^3^ The patient noticed a reduction in treatment efficacy although she was injected with similar doses as previous treatments, and that treatment effects lasted for a shorter time compared with previous treatments; ^4^ The patient experienced no clinical effect for all treatments, indicating complete non-response to BoNT-A.

**Figure 3 toxins-15-00456-f003:**
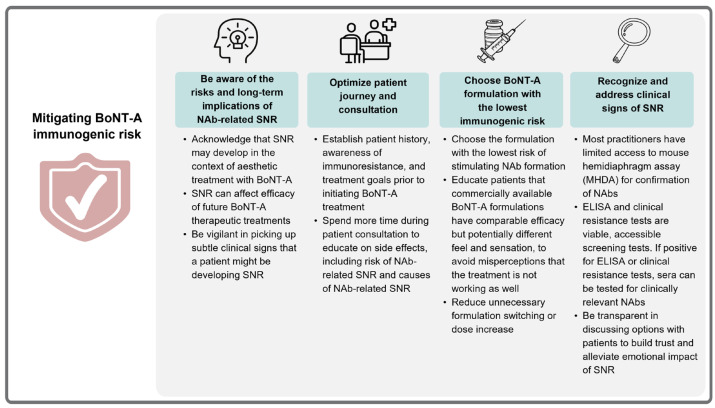
Preferred strategies to mitigate immunogenic risk with BoNT-A treatment. BoNT-A: botulinum neurotoxin A; NAb: neutralizing antibody; SNR: secondary non-response.

## Data Availability

Data available upon reasonable request; queries should be directed to the corresponding author.

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
