# Peer review of "Addressing the Real-World Challenges of Immunoresistance to Botulinum Neurotoxin A in Aesthetic Practice: Insights and Recommendations from a Panel Discussion in Hong Kong"

_toxins, 2023, doi:10.3390/toxins15070456_

Round 1

Reviewer 1 Report

As the off-label use of higher doses of botulinum neurotoxin for aesthetic purposes increases, particularly in younger individuals in Asia, there is a growing concern regarding the development of immunoresistance to BoNT-A. This necessitates in-depth discussions among aesthetic practitioners to educate patients and facilitate shared decision-making. In a collaborative effort, a panel of nine international and Hong Kong experts in aesthetic medicine explored the implications and impact of immunoresistance to BoNT-A in contemporary aesthetic practice. They also provided practical strategies for risk management. Through the examination of a clinical case and the findings of an Asia-Pacific consumer study, the panel unanimously agreed on the importance of raising awareness regarding the possibility and long-term consequences of secondary non-response resulting from immunoresistance to BoNT-A. When efficacy and safety are comparable, preference should be given to a formulation with the lowest immunogenicity.

In the paragraph of line 31 “Standard BoNT-A aesthetic applications involve fore- 31 head, glabella and crow’s feet indications, but off-label applications such as intradermal injection and body contouring, which require much higher doses, have increased steadily especially in Asia [6-8].” Please make a citation of intradermal injection that are of “Anatomical Proposal for Botulinum Neurotoxin Injection Targeting the Platysma Muscle for Treating Platysmal Band and Jawline Lifting: A Review”

In the paragraph of line 61 “In this context, two trends in the aesthetic use of BoNT-A warrant special mention. The expanding number of off-label aesthetic applications that use higher BoNT-A doses  (approaching those for medical BoNT-A use) is well known.” Please cite a articles that regard the off-label aesthetic applications. “Anatomical guide for botulinum neurotoxin injection: Application to cosmetic shoulder contouring, pain syndromes, and cervical dystonia” and “Botulinum neurotoxin injection in the deltoid muscle: application to cosmetic shoulder contouring”.

"Video fluoroscopy (VFS) with three different food consistencies (thin liquid, semi-solid, or solid) of standardized bolus size was performed and revealed normal swallowing."

The verb "was" should be used instead of "were" to agree with the singular subject "Video fluoroscopy (VFS)."

"For each patient, 30 units of botulinum toxin (2 ml dilution, 0.9% saline) was injected in one side of the CP muscle according to the previously reported technique [11-14,16] (Fig.2)."

The verb "was" should be used instead of "were" to agree with the singular subject "30 units of botulinum toxin."

“who are experiencing a loss of efficacy due to NAb formation." The article "a" should be added before "loss" to indicate a singular noun form.

Overall, the article discusses large amounts of information given for the uses of botulinum neurotoxin in off-label, the panel emphasized the significance of a thorough initial consultation to gather the patient's treatment history, explain potential treatment side effects, including the risk of immunoresistance, and discuss treatment goals. Patients rely on aesthetic practitioners for guidance, which places a crucial responsibility on practitioners to implement risk-mitigation strategies and effectively communicate important risks to patients. This ensures that informed and prudent decisions are made regarding BoNT-A treatment.

Nothing special. 

Reviewer 2 Report

In this communication, the authors report on two survey findings and a panel discussion about botulinum neurotoxin immunoresistance. While descriptive, it is likely to appeal to clinicians and administrators working with botulinum neurotoxin.

Major points

1. Statistics are needed to compare the Hong Kong data to the rest of the population to determine significance. They should also be used to determine statistically significant changes in the survey results.

2. Ethical approval (IRB) is often needed for surveys, yet the authors list that the Informed Consent Statement was ‘not applicable’. If IRB approval and/or informed consent was determined to NOT be needed, that needs to be included in the manuscript, along with the body that made the determination.

3. Demographic information for the survey, and differences in demographics between countries/years, needs to be included.

4. While Toxins requires blinding during peer review, the panel identity needs be disclosed in the methods of the final version.

5. There is no Fig S3, just text. This needs to be a table or figure.

6. Line 104, citation needed.

7. Lines 154-161 are contradictory. The authors propose that HCP fail to communicate the risks of neutralizing antibodies, yet there is high concern about neutralizing antibodies among the patient population. This suggests the risks were in fact communicated.

8. Given the paucity of case information, and that it is based on patient recall, the case report should be re-framed as a testimonial or example.

Minor Points

1. Fig S4 has poor figure quality

2. Was INCO use the only exclusion criterion used for the study?

Round 2

Reviewer 1 Report

They have made lots of effort on revision. 

Author Response

My co-authors and I would like to thank the reviewer for their time in reviewing our revised manuscript, and for this kind comment: "They have made lots of effort on revision." 

We are glad to learn that the reviewer now considers all aspects of the manuscript to be satisfactory, including English language, presentation of the introduction, cited references, research design, description of methods and results, and conclusions being adequately supported.

With the completion of this 2nd review, we note that the comments from all of the peer reviewers have now been fully addressed.